# Estimating the effects of meteorology and land cover on fire growth in Peru using a novel difference equation model

Harry Podschwit[1], William Jolly[1], Ernesto Alvarado[2], Andrea Markos[3], Satyam Verma[2], Sebastian Barreto-Rivera[3], Catherine Tobón-Cruz[3], and Blanca Ponce-Vigo[3]

[1]US Forest Service, Rocky Mountain Research Station, Fire Sciences Laboratory, 5775 Highway 10 West, Missoula, MT, USA
[2]School of Environmental and Forest Sciences, University of Washington, Seattle, USA
[3]US Forest Service, International Programs, 1 Thomas Circle, NW Suite 400 Washington D.C., U.S.A.

**Correspondence:** Harry Podschwit (hpodschwit@gmail.com)

**Abstract.** Statistical analyses of wildfire growth are rarely undertaken, particularly in South America. In this study, we describe a simple and intuitive difference equation model of wildfire growth that uses a spread parameter to control the radial speed of the modeled fire and an extinguish parameter to control the rate at which the burning perimeter becomes inactive. Using data from the GlobFire project, we estimate these two parameters for 1003 large, multi-day fires in Peru between 2001 and 2020. For four fire-prone ecoregions within Peru, a set of 24 generalized linear models are fit for each parameter that use fire danger indexes and land cover covariates. Akaike weights are used to identify the best-approximating model and quantify model uncertainty. We find that, in most cases, increased spread rates and extinguish rates are positively associated with fire danger indexes. When fire danger indexes are included in the models, the spread component is usually the best choice, but also found instances when the fire weather index and burning index were selected. We also find that grassland cover is positively associated with spread rates and extinguish rates in tropical forests, and that anthropogenic cover is negatively associated with spread rates in xeric ecoregions. We explore potential applications of this model to wildfire risk assessment and burned area forecasting.

## 1 Introduction

Although researchers frequently characterize fire in terms of size (Doerr and Santín, 2016), there are a number of other parameters that describe unique dimensions of fire behavior. Spread rates, for instance, are an important parameter for understanding fire entrapment and burnover risks. When spread rates are low, nearby individuals have more time to detect and escape an advancing fire front than when spread rates are high (Page et al., 2019). Indeed, unexpected and explosive fire growth has been implicated in a large number of fatal wildfires (Viegas and Simeoni, 2011) and rarely prescribed fire (Twidwell et al., 2015) accidents. For this reason, numerous models (Sullivan, 2009b, a), policies (Butler, 2014), technologies (De Vivo et al., 2021), and tools (Jolly et al., 2019) have been developed to mitigate the risk of high-spread events to firefighters and the general public. Fire spread rates are also an important parameter for understanding firefighting effectiveness (Rapp et al., 2021; Finney et al., 2009), and periods of low spread are often good opportunities for safe and productive fire suppression. Like fire spread, descrip-

tions of how and when the length of the burning perimeters reduce over time can be valuable to decision makers. Much of the research deals with this parameter indirectly by focusing on extinguishment in terms of duration (Andela et al., 2019) or fireline production rates (Fried and Fried, 1996). However, the length of burning perimeter may be reduced for factors unrelated to fire suppression. For instance, the burning perimeter may be reduced when a portion of the fire front spreads into a preexisting fuel break such as previously burned areas, waterbodies, roads, and inflammable vegetation (Reed and McKelvey, 2002). Similarly, local weather may differentially control fire spread and extinguishment, leading to scenarios where one portion of a fire may be spreading at the same time another portion is dying out (Price et al., 2014; Reed and McKelvey, 2002). Understanding of the dynamics and relevant factors mediating fire extinguishment can improve fire-related decisions through multiple means such as reducing redundancies in fireline construction strategies (Wei et al., 2021), and improving the cost-effectiveness of firefighting (Houtman et al., 2013).

At least part of the reason that fire parameters such as spread and extinguish rates are overlooked by researchers is that the relevant data are uncommon. Daily fire growth data are rare, usually coming from case studies and administrative records (Taylor et al., 2013), and although fire perimeter data are occasionally available (Zhong et al., 2016), identifying which portions of the perimeter are active and inactive can be a difficult task (Anderson et al., 2009). Where these data are available, they are subject to fairly high levels of uncertainty and often contain gaps and errors (Podschwit et al., 2018; Kolden and Weisberg, 2007). These problems are particularly noticeable in locations that lack the historical records and technological resources that are more commonly found in the United States, Canada, and Australia. In South America for example, data regarding fire spread rates are available from only a few experimental studies (Ray et al., 2005; Bufacchi et al., 2017), which may not be representative of the real-world conditions (Melcher et al., 2016) nor reliably be extrapolated to other locations. In most cases, information about real-world fire spread and extinguishment must be derived from satellite data (Andela et al., 2019).

In spite of these problems with data quantity and quality, we can use available research to develop some intuition regarding which environmental factors are likely to be relevant to Peruvian fire spread and extinguishment. Firefighter entrapments in the United States have been typically associated with rapid fire spread (Page et al., 2019), and this fire parameter is both directly (Rapp et al., 2021) and indirectly (Jolly et al., 2019) associated with low fuel moisture and anomalously high wind speed. Consequently, a number of indexes have been proposed to estimate the risk of rapidly spreading fire based on the relevant atmospheric conditions. The energy release component (ERC) and burning index (BI) are commonly used in the United States to inform firefighting decisions (Jolly et al., 2015; Cullen et al., 2020) and are calculated from a complex equation of tempera-ture, precipitation, wind, humidity, cloud cover, fuel, topographic, and geographic data (Deeming et al., 1977; Bradshaw et al., 1984). The spread component (SC) is derived from ERC and BI, and provides measure of idealized fire spread in certain fuel conditions (Bradshaw et al., 1984). The fire weather index (FWI) was developed in the 1970's and is the preferred choice of FDI in Canada (Van Wagner et al., 1974; Bradshaw et al., 1984) and is structured somewhat similarly to the BI (Fujioka et al., 2008). The Keetch-Byram Drought Index (KBDI) measures the water balance of the upper soil layers (Littell et al., 2016; Keetch and Byram, 1968) and was developed to predict forest fire activity in the Southeastern United States. Although devel-oped for other regions, these FDIs can sometimes be informative of fire activity in other locations (Podschwit et al., 2022). Like FDIs, we can also develop some intuition about the effects of landcover on fire behavior using information from other

locations. For instance, it is well known that vegetation can influence fire growth through sheltering of the flames from wind (Massman et al., 2017) and we might predict that forested land cover would have slower fire spread than non-forested land cover. It has also been shown that human presence, and by extension anthropogenic land cover types, can sometimes have an inhibitory effect on various fire parameters, although the relationship is complex and non-monotonic (Bistinas et al., 2013).

Existing conceptual climate-fire models can also help us predict how fuel availability and flammability are likely to mediate fire spread in novel environments. We call ecosystems that have abundant fuel available (e.g. forests) climate-limited, and these ecosystems typically require exceptionally dry periods to permit large fire growth. On the other hand, we call ecosystems that are frequently dry typically have low levels of fuel (e.g. grasslands) fuel-limited. Unlike climate-limited ecosystems, fuel-limited ecosystems require above-average antecedent precipitation to produce fuel loads of sufficient quantity and continuity to permit large fire growth (Meyn et al., 2007). Indeed, naturally occurring large fires in the Amazon - an unambiguously climate-limited ecosystem - are very rare (Lima et al., 2012) and require extremely low and sustained moisture to permit combustion (Cochrane, 2003). When fire does occur in the Amazon, they are usually intentionally set to clear forest for agriculture (Cochrane, 2003) and are characterized by low spread rates ($\ll 1$ m/min) and low flame height ($< 0.5$ m) (Cochrane, 2003; Ray et al., 2005; Bufacchi et al., 2017). In the Peruvian Andes - an ecoregion that is largely unforested - precipitation patterns follow a sawtooth pattern during peak fire years, where precipitation is anomalously high in the year previous, followed by dry weather during the peak fire year (All et al., 2017). Similarly, ENSO-related increases in precipitation in the Sechura desert of northern Peru are associated with increased fuel loads and subsequent increases in fire activity (Block et al., 2000). Like conceptual models of fire activity, satellite-derived estimates of fire spread can also provide intuition regarding the scale and variability of fire spread in Peru. In South America, the typical individual fire is reported to increase in area at a rate of 0.5 km$^2$ per day and increase radially at a rate of radial spread estimates of 0.7-0.8 km per day. These growth estimates do not appear to strongly vary within South America, but large differences are found between xeric regions versus humid tropical regions globally (Andela et al., 2019).

Given that meteorology and land cover are known to be important influences on fire spread in other locations, we would like to explore the relevance of these factors to fire spread in Peru, where these relationships have not been well quantified. To do this, we will develop a simple difference equation model to summarize satellite-derived fire growth data. We will then build regression models to predict the difference equation parameters and determine if a suite of meteorological and land cover covariates are statistically significant predictors. The intended outcome of this analysis is to present a relatively simple method of predicting fire growth based on environmental conditions that can be applied nearly anywhere globally.

## 2   Model

The difference equation model begins with two simplifying assumptions regarding fire growth:

1. Fire spreads at a constant rate, $r$, from all angles from an ignition point.

2. After the first time step, a constant length, $l$, of the fire's perimeter is extinguished.

A graphical description of these dynamics for various values of $r$ and $l$ is shown below (Figure 1). Note that in this study the word "growth" is reserved to describe changes in fire area and "spread" is reserved to describe radial changes in fire.

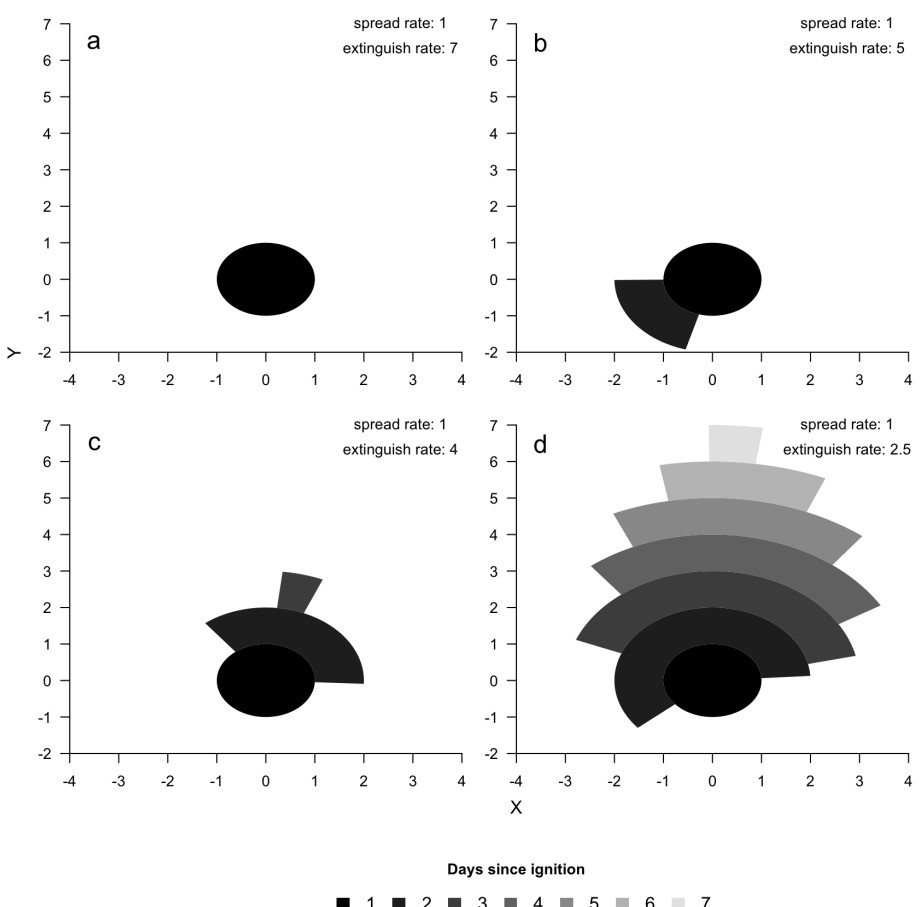

**Figure 1.** Graphical depiction of modeled fire growth for four pairs of spread rate and extinguish rate parameters. The X and Y dimension represent the distance from the ignition point at an arbitrary scale.

## 2.1 Sector length and arc

We hereafter assume that $r$ and $l$ are measured in kilometers per day. It follows from these assumptions that on the first day, the fire is a circle with a radius of $r$ km and $2\pi r$ km of initial burning perimeter. There is no extinguishment of burning perimeter on the first day. On the second day the amount of burning perimeter is reduced by $l$ km and the fire continues to spread radially at rate $r$ km/day. For any $l \geq 2\pi r$, there is no fire growth on the second day since all of the burning perimeter is extinguished. The sector arc of burning perimeter at the start of the second day is calculated via (Equation 1).

$$\theta_2 = \frac{\max\left(2\pi r - l, 0\right)}{r}. \tag{1}$$

The second day's final burning perimeter length is calculated from (Equation 2)

$$S_2 = \theta_2 \times (2r); \tag{2}$$

and in general, we can calculate the current burning perimeter length using (Equation 3).

$$S_t = \theta_t \times (tr). \tag{3}$$

Moreover, since the sector arc is calculated by taking the sector length and dividing by the radius, we can see that the sector arc follows (Equation 4).

$$\theta_t = \frac{\max\left(S_{t-1} - l, 0\right)}{r(t-1)}. \tag{4}$$

We can now use (Equation 3) to greatly simplify (Equation 4). Namely, if $t > 1$, then

$$\theta_t = \frac{\max\left(\theta_{t-1}(t-1)r - l, 0\right)}{r(t-1)}. \tag{5}$$

If we assume that the fire is not going to be extinguished at time $t$, then $S_{t-1} \geq l$ and (Equation 5) may be further simplified into the following difference equation (Equation 6).

$$\theta_t = \theta_{t-1} - \frac{l}{r(t-1)}. \tag{6}$$

An exact solution to the difference equation is produced by noting,

$$\Delta\theta_t = \theta_t - \theta_{t-1} = -\frac{l}{r(t-1)}. \tag{7}$$

And that for $t > 1$,

$$\theta_t = \theta_1 + \sum_{n=2}^{t} \Delta\theta_n = \theta_1 - \frac{l}{r}\sum_{n=2}^{t}\frac{1}{(n-1)}. \tag{8}$$

The final equation of the sector arc dynamics (Equation 9) is produced by noting the presence of a harmonic series, $H_{t-1}$.

$$\theta_t = \theta_1 - \frac{l}{r}H_{t-1}. \tag{9}$$

## 2.2 Area and duration

The equation representing the burning perimeter can be manipulated to represent more commonly used wildfire parameters such as area and duration. Cumulative burn area over time can be calculated by noting that each day's growth can be represented as a partial annulus (Figure 1, Equation 10) and taking the sum of these daily growth predictions (Equation 11).

$$\Delta A_t = A_t - A_{t-1} = \frac{\theta_{t-1}}{2} \times [(t \times r)^2 - ((t-1) \times r)^2]. \tag{10}$$

$$A_t = \sum_{n=1}^{\infty} \Delta A_t = \frac{r^2}{2} \sum_{n=1}^{\infty} \theta_{n-1} \times (n^2 - (n-1)^2) = \frac{r^2}{2} \sum_{n=1}^{\infty} \theta_{n-1} \times (2n-1). \tag{11}$$

In practice, only a partial series is required because the fire will cease to grow at some point in time. The number of days from ignition until wildfire growth ceases can be derived from (Equation 9) and we can see that the modeled fire will grow for all $t$ such that $\frac{\theta_1 r}{l} > H_{t-1}$.

## 2.3 Estimation of model parameters from area and duration

Note in (Equation 9) that if the day 1 burning arc, $\theta_1$, is known, then the dynamics of the burning arc, and by extension the duration. are entirely determined by the ratio of the spread and extinguish rates, $\gamma = \frac{l}{r}$, a quantity we hereafter refer to as the relative decay rate. Hence, if a burned area time series with area ($A$) and duration ($T$) was desired, then estimates of the spread rate $\hat{r}$ and extinguish rate $\hat{l}$ could be obtained by choosing a $\hat{\gamma}$ such that,

$$\frac{\theta_1}{H_T} \leq \hat{\gamma} < \frac{\theta_1}{H_{T-1}}. \tag{12}$$

Next, a normalizing factor, $c$, is estimated (Equation 12) that assumes a spread rate of $r = 1$ and produces a fire progression with a known duration via the relative decay rate, $\hat{\gamma}$.

$$c = \frac{1}{2} \sum_{n=1}^{T} \theta_{n-1|\gamma=\hat{\gamma}} \times (2n-1). \tag{13}$$

An estimate of $\hat{r}$ that produces the desired final area can then be obtained by using (Equation 11),

$$\hat{r} = \sqrt{\frac{A}{c}}. \tag{14}$$

The difference equation produced from $\hat{r}$ and $\hat{l} = \hat{r}\hat{\gamma}$ will have duration $T$ and final area $A$.

## 3  Application

### 3.1  Data and preprocessing

GlobFire data (Artés et al., 2019) were clipped to boundaries of Peru, and disaggregated into four ecoregions as defined from the 2009 Nature Conservancy assessment [1]. Each ecoregion is approximately similar in terms of climate and vegetation. Regional variability in fire detection probabilities were lessened by limiting our study to large and long duration events, which are likely to be detected regardless of forest cover and atmospheric conditions. Hence, incidents that were smaller than 405 hectares or incidents that reportedly burned for one day were removed from the main analysis. The spatially explicit growth

maps produced from GlobFire were converted into spatially-inexplicit burned area time series and the centroid of the final perimeter recorded. Summary statistics of the incidents disaggregated by ecoregion are reported in (Table 1) and a map of the incident locations is shown in (Figure 2).

**Table 1.** Regional sample size plus quartiles for both fire size (km$^2$) and duration (days). The summary statistics calculated from the data without size-based filtering are reported in the parenthesis.

| Ecoregion | Sample size | Size | | | | | Duration | | | | |
|---|---|---|---|---|---|---|---|---|---|---|---|
| | | $Min$ | $Q_1$ | $Q_2$ | $Q_3$ | $Max$ | $Min$ | $Q_1$ | $Q_2$ | $Q_3$ | $Max$ |
| Xeric | 38 (342) | 4.13 (0.41) | 4.81 (0.92) | 5.96 (1.41) | 9.51 (2.54) | 65.70 | 2 (2) | 5 (3) | 8 (5) | 10 (7) | 21 (21) |
| Andean | 252 (1395) | 4.11 (0.41) | 4.83 (0.70) | 6.49 (1.40) | 10.76 (2.98) | 69.33 | 2 (2) | 5 (3) | 6 (4) | 9 (6) | 24 (24) |
| Dry forest | 50 (277) | 4.25 (0.43) | 5.01 (0.71) | 7.09 (1.42) | 14.50 (3.31) | 289.44 | 3 (2) | 5 (3) | 7 (5) | 10.75 (7) | 18 (18) |
| Amazon | 663 (5772) | 4.14 (0.41) | 4.88 (0.70) | 6.34 (1.16) | 9.05 (2.12) | 91.52 | 2 (2) | 7 (3) | 9 (5) | 13 (7) | 40 (40) |

Five fire danger indexes (FDIs) are calculated using ERA5 reanalysis for 2001-2020[2]: the burning index (BI), energy release component (ERC), spread component (SC) (Bradshaw et al., 1984), fire weather index (FWI), and the Keetch-Byram

Drought Index (KBDI) (Littell et al., 2016). The raw values were converted into a score by subtracting the mean and dividing by the standard deviation calculated from 2001-2020 data. Additionally, land cover data came from the GlobCover dataset (Arino et al., 2007) [3]. GlobCover data uses satellite measurements to classify land cover globally into one of 22 categories at a 300 meter resolution. GlobCover data were used to characterize the general composition of land cover within the burned area of wildfires.

The FDI values for each incident were extracted on the reported ignition date of the fire and at the incident centroid. The number of pixels within the final burn perimeter from each land cover type was recorded, reclassified using (Table 2), and converted into percentages.

---

[1]https://geospatial.tnc.org/datasets/b1636d640ede4d6ca8f5e369f2dc368b/about
[2]https://www.wfas.net/data/SAR/
[3]http://due.esrin.esa.int/page_globcover.php

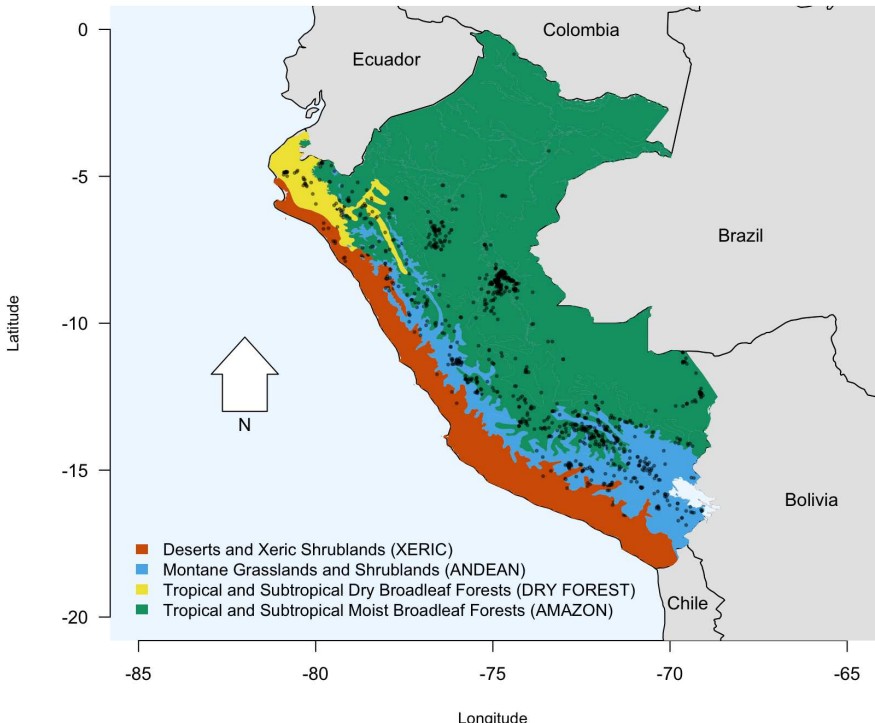

**Figure 2.** Map of the Peruvian ecoregions considered in this study with 2001-2020 GlobFire centroids of large, multi-day fires overlaid.

### 3.2 Estimation of model parameters

For each incident, the spread rate ($r$) and the extinguish rate ($l$) are estimated using the burned area time series derived from GlobFire. Specifically, for each burned area time series, the values of $r$ and $l$ that minimized the root mean square error (RMSE) were identified using the Nelder-Mead optimization algorithm (Nelder and Mead, 1965). To increase the likelihood that the global optimum was identified, 1000 initial best guesses of the model parameters were obtained by using a normal distribution with standard deviation one to jitter the estimates produced with the methods described in section 2.3. The optimization routine was run for each of the 1000 best guesses and the run producing the lowest root mean square error was assumed to be the global optimum. The correlation of these two parameters was explored using two methods. Firstly, the spearman correlations of the spread rate and extinguish rates estimates were calculated for each ecoregion. Secondly, a quadratic model that predicted the extinguish rate from the spread rate was fit using least-squares regression.

### 3.3 Statistical modeling

An initial set of 24 generalized linear models were considered for each ecoregion and model parameter. The 24 candidates were produced by considering all linear combinations of FDIs and land cover covariates such that at most one variable from each

**Table 2.** Reclassification of GlobCover data

| GlobCover value | New value |
|---|---|
| Post-flooding or irrigated croplands (or aquatic) | Anthropogenic |
| Rainfed croplands | |
| Mosaic cropland (50-70%) vegetation (grassland/shrubland/forest) (20-50%) | |
| Mosaic vegetation (grassland/shrubland/forest) (50-70%) / cropland (20-50%) | |
| Artificial surfaces and associated areas (Urban areas >50%) | |
| Closed to open (>15%) broadleaved evergreen or semi-deciduous forest (>5m) | Forest |
| Closed (>40%) broadleaved deciduous forest (>5m) | |
| Open (15-40%) broadleaved deciduous forest/woodland (>5m) | |
| Closed (>40%) needleleaved evergreen forest (>5m) | |
| Open (15-40%) needleleaved deciduous or evergreen forest (>5m) | |
| Closed to open (>15%) mixed broadleaved and needleleaved forest (>5m) | |
| Mosaic forest or shrubland (50-70%) | |
| Closed to open (>15%) broadleaved forest regularly flooded | |
| Closed (>40%) broadleaved forest or shrubland permanently flooded | |
| Closed to open (>15%) grassland or woody vegetation on regularly flooded or waterlogged soil | |
| Mosaic grassland (50-70%) | Grassland |
| Closed to open (>15%) shrubland (<5m) | |
| Closed to open (>15%) herbaceous vegetation | |
| Sparse (<15%) vegetation | |

variable category was used. The generalized linear model assumed a inverse-link and Gamma density function. The Gamma distribution was selected because it, like the model parameters, is defined over a semi-infinite support, and when compared to the log-normal or inverse-Gaussian distribution, provided a superior goodness-of-fit. The goodness-of-fit of the Gamma density function[4] was verified via visual inspection of Q-Q plots with a 95% simultaneous confidence band (see Appendix A) and was calculated using the *distrMod* package (R Core Team, 2013). The inverse function was selected because it is the canonical link function for the Gamma distribution (Faraway, 2016). The best models were selected using Akaike weights (Wagenmakers and Farrell, 2004), which represent the probability that a model candidate was the true best-approximating model from the original set of 24 models. Akaike weights are calculated from a commonly used model selection criterion, Akaike information criterion (AIC), but has the added benefit of gauging the level of model uncertainty with probabilities, which are easier to interpret than the raw AIC values. The out-of-sample performance of the best-approximating models were estimate using a 3-fold cross-validation. Each fire event for each region was placed into one of three folds according to the year in which the fire event occurred: 2000-2006, 2007-2013, and 2014-2020. Each fold was used as a test set once and used

---

[4]Q-Q plots of the lognormal and inverse-gamma distribution were omitted to keep the manuscript concise, but can be requested from the corresponding author.

as a training set twice (Hastie et al., 2009). The average RMSE (Equation 15) and symmetrical mean absolute percent error (Equation 16) was used to measure out-of-sample model performance. All calculations of this analysis were performed in the R programming environment (R Core Team, 2013).

$$RMSE = \sqrt{\frac{1}{n}\sum_{i=1}^{n}(y_i - f_i)^2} \tag{15}$$

$$sMAPE = \frac{200}{n}\sum_{i=1}^{n}\frac{|y_i - f_i|}{y_i + f_i} \tag{16}$$

### 3.4 Sensitivity analysis

Because size-based filtering of wildfire events can effect the results of statistical analyses (Podschwit and Cullen, 2020), the main analysis was also repeated without filtering and compared to the original analysis to describe the consequences of this methodological choice. Specifically, changes to the structure of the best-approximating model and changes in Akaike weights were identified.

## 4 Results

### 4.1 Difference equation parameters and performance

Overall, mean fire spread rates were approximately a half a kilometer per day, but this quantity varied by ecoregion. The mean spread rates were highest in the Andean and dry forest ecoregions, and was lowest in the Amazon ecoregion. The mean extinguish rate were usually near two kilometers per day, but like spread rates, this quantity varied by ecoregion. The mean extinguish rate was highest in the dry forest ecoregion and lowest in the Amazon ecoregion (Figure 3). The gamma distribution provided a high-quality parametric approximation of the distribution of observed spread rates (Figure A1) and a slightly lower quality approximation of the distribution of extinguish rates (Figure B1).

The spread rate and extinguish rate were highly correlated with one another. Spearman correlation coefficients ranged from 0.91 in the dry forest ecoregion to 0.96 in the Amazon and Xeric ecoregions. The observed relationship between the two parameters was well-described using a quadratic model (Figure 4).

In both absolute and relative terms, the difference equations well-approximated the observed burned area time series across all ecoregions. The lowest median RMSE ($0.54\,\text{km}^2$) was seen in xeric ecoregion, whereas the highest median RMSE was seen in dry forests ($0.76\,\text{km}^2$). Similarly, the lowest sMAPE was observed in the xeric ecoregion (20.36 percent) and the highest sMAPE was observed in the dry forests (23.25 percent). In terms of RMSE, the difference equation model did particularly well on fires smaller than $10\,\text{km}^2$ and fires less than three weeks in duration (Figure 5). However, when sMAPE was instead considered, there was little difference in performance between small short-duration fires and larger, long duration fires.

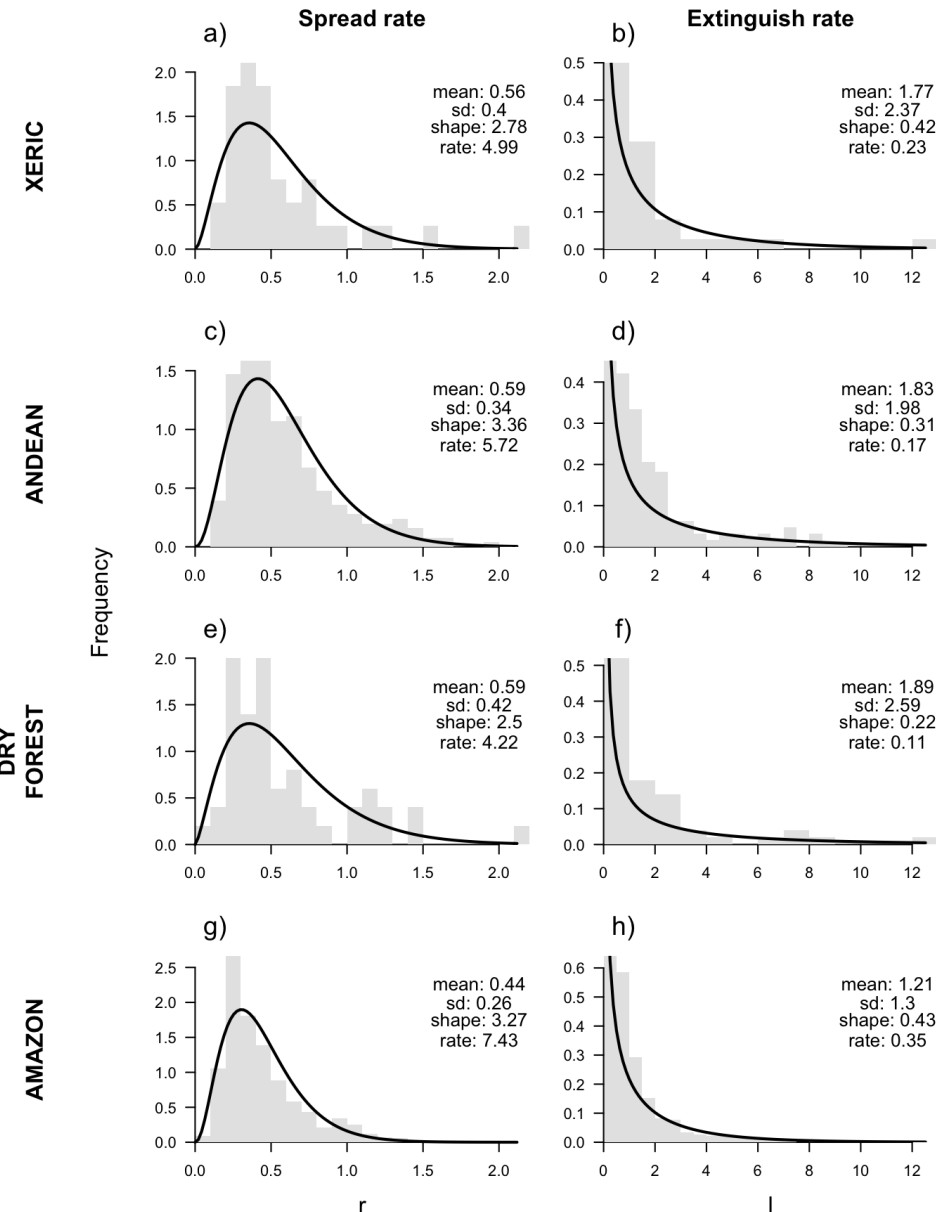

**Figure 3.** Histograms of the spread rates and extinguish rates (in kilometers per day) disaggregated by ecoregion. Maximum-likelihood estimates of the gamma distribution are overlaid. The mean, standard deviation, estimated shape, and estimated rate parameters are reported in the upper right of each panel.

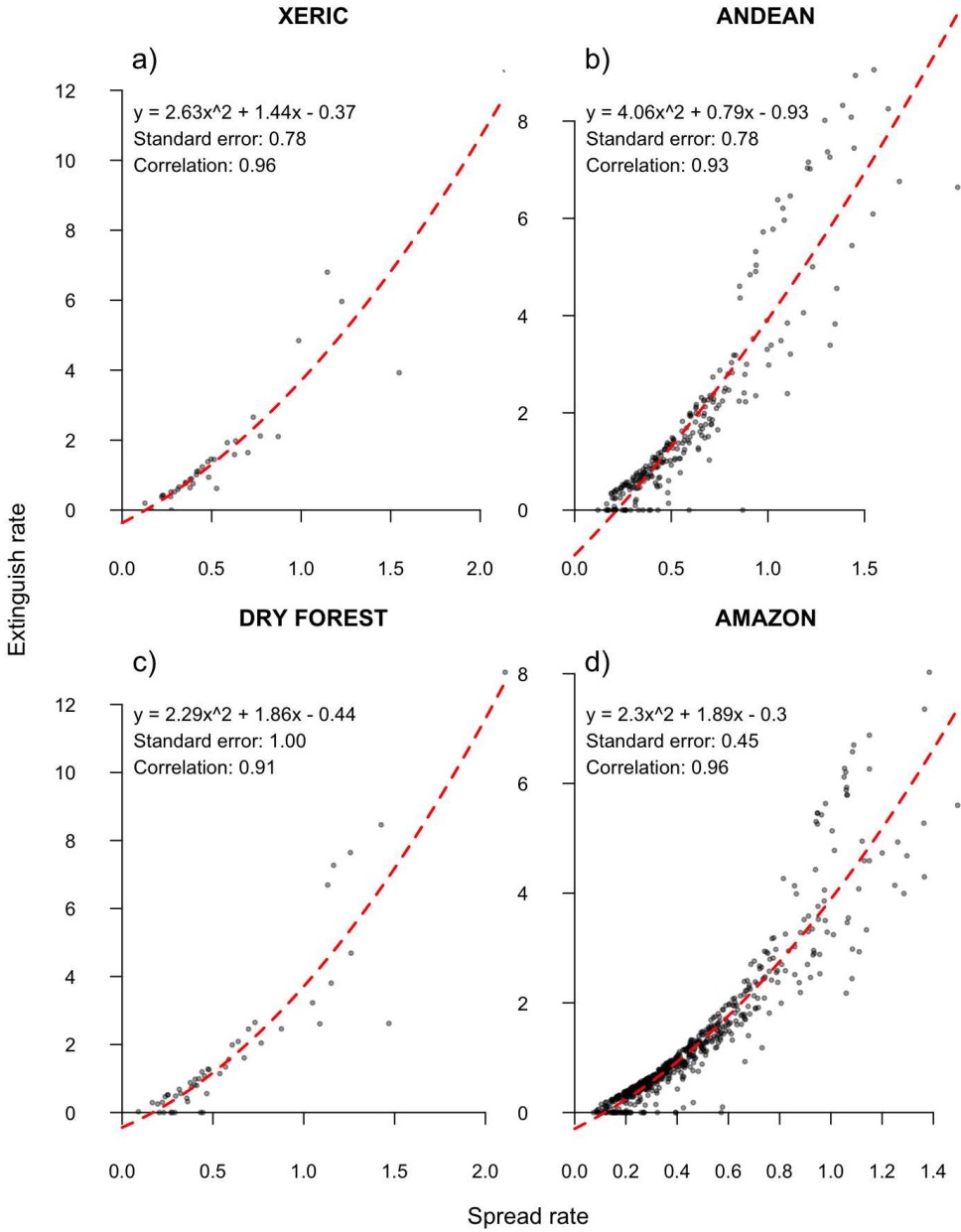

**Figure 4.** Scatterplots of observed relationship between spread rate and extinguish rate (in kilometers) disaggregated by ecoregion. Least-squares fit of quadratic model and spearman correlation coefficient are reported in upper left of each panel.

## 4.2 Best-fitting parameter models

In the xeric ecoregion, the best-approximating model of spread rates predicted that fire would grow faster as FWI increased and as the percent of anthropogenic cover decreased, but the relationships were only weakly statistically significant. In the

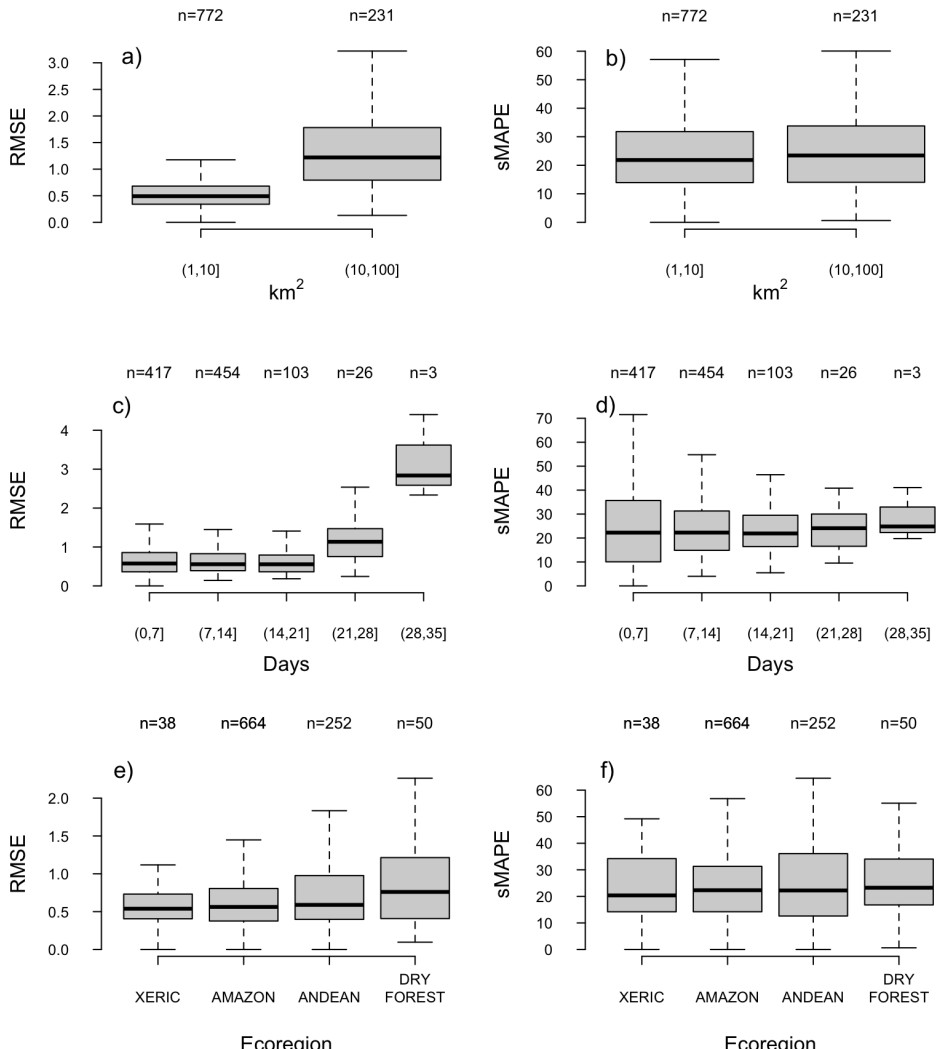

**Figure 5.** Boxplots of root-mean square error estimates calculated from best-fit difference equation predictions and GlobFire observations. Root-mean square error estimates are disaggregated by size, duration, and ecoregion. Sample size is reported above the boxplots and outliers have been removed for visualization purposes.

Andean ecoregion, a strongly significant positive relationship between fire spread rates and the SC was observed. The best-approximating spread model in the dry forest ecoregion did not include any FDI or land cover variables. In the Amazon ecoregion, the best-approximating model of spread assumed a positive relationship with SC and a positive relationship with percent grassland land cover. In the Amazon ecoregion, both the effect of the SC on fire spread and the effect of landcover of fire spread was very-strongly significant (Table 3).

**Table 3.** Summary of spread parameter generalized linear models for each ecoregion. Summary including formula, Akaike weight, and significance level. P-values greater than 0.10 are interpreted as not significant (ns), P-values less than 0.1 are interpreted as weakly significant (.), P-values less than 0.05 are interpreted as significant (\*), P-values less than 0.01 are interpreted as strongly significant (\*\*), and P-values less than 0.001 are interpreted as very-strongly significant (\*\*\*).

| Ecoregion | $-\frac{1}{\mu}$ | Dispersion | AIC weight | P-value | |
| --- | --- | --- | --- | --- | --- |
| | | | | FDI | Land cover |
| XERIC | $-1.87 + 0.20 \times FWI - 1.85 \times percent.anthro$ | 0.43 | 0.17 | . | . |
| ANDEAN | $-1.87 + 0.17 \times SC$ | 0.33 | 0.23 | \*\* | |
| DRY FOREST | $-1.68$ | 0.50 | 0.11 | | |
| AMAZON | $-2.86 + 0.14 \times SC + 1.05 \times percent.grass$ | 0.30 | 0.89 | \*\*\* | \*\*\* |

In the xeric and dry forest ecoregions, the extinguish rate models did not include any covariates. In the Andean ecoregion, SC had a strongly significant and positive relationship with the extinguish rate. In the Amazon ecoregion, extinguish rates were predicted to increase with the SC and percent grassland land cover; both covariates were very-strongly significant (Table 4).

**Table 4.** Summary of extinguish rate generalized linear models for each ecoregion. Summary including formula, Akaike weight, and significance level. P-values greater than 0.10 are interpreted as not significant (ns), P-values less than 0.1 are interpreted as weakly significant (.), P-values less than 0.05 are interpreted as significant (\*), P-values less than 0.01 are interpreted as strongly significant (\*\*), and P-values less than 0.001 are interpreted as very-strongly significant (\*\*\*).

| Ecoregion | $-\frac{1}{\mu}$ | Dispersion | AIC weight | P-value | |
| --- | --- | --- | --- | --- | --- |
| | | | | FDI | Land cover |
| XERIC | $-0.56$ | 1.79 | 0.10 | | |
| ANDEAN | $-0.64 + 0.09 \times SC$ | 1.16 | 0.11 | \*\* | |
| DRY FOREST | $-0.53$ | 1.87 | 0.15 | | |
| AMAZON | $-1.15 + 0.07 \times SC + 0.53 \times percent.grass$ | 1.09 | 0.35 | \*\*\* | \*\*\* |

For the spread models, the Akaike weights - the probability that the true best-approximating model was selected from the 24 candidates - ranged from 0.11 in the dry forest ecoregion to 0.89 in the Amazon ecoregion (Table 3). For the extinguish rate models, this probability ranged from 0.10 in the xeric ecoregion to 0.35 in the Amazon ecoregion (Table 4). The probability that the true best-approximating spread model contained both a land cover and FDI covariate ranged was 0.43 in the dry forest ecoregion, 0.54 in the Andean ecoregion, 0.68 in the xeric ecoregion, and >0.99 in the Amazon. The probability that the true best-approximating extinguish model contained both a land cover and FDI covariate was lower, and ranged from 0.36 in the dry forest ecoregion, 0.45 in the Andean ecoregion, 0.47 in the xeric ecoregion, and 0.88 in the Amazon ecoregion (Figure 6).

The spread rate estimates produced from the best-approximating generalized linear models were expected to get within about 0.3 to 0.4 of the best-fitting spread rate estimates, but this performance varied by ecoregion. The best model performance was seen in the Amazon ecoregion, and the worst model performance was seen in the Dry forest ecoregion, but the variability

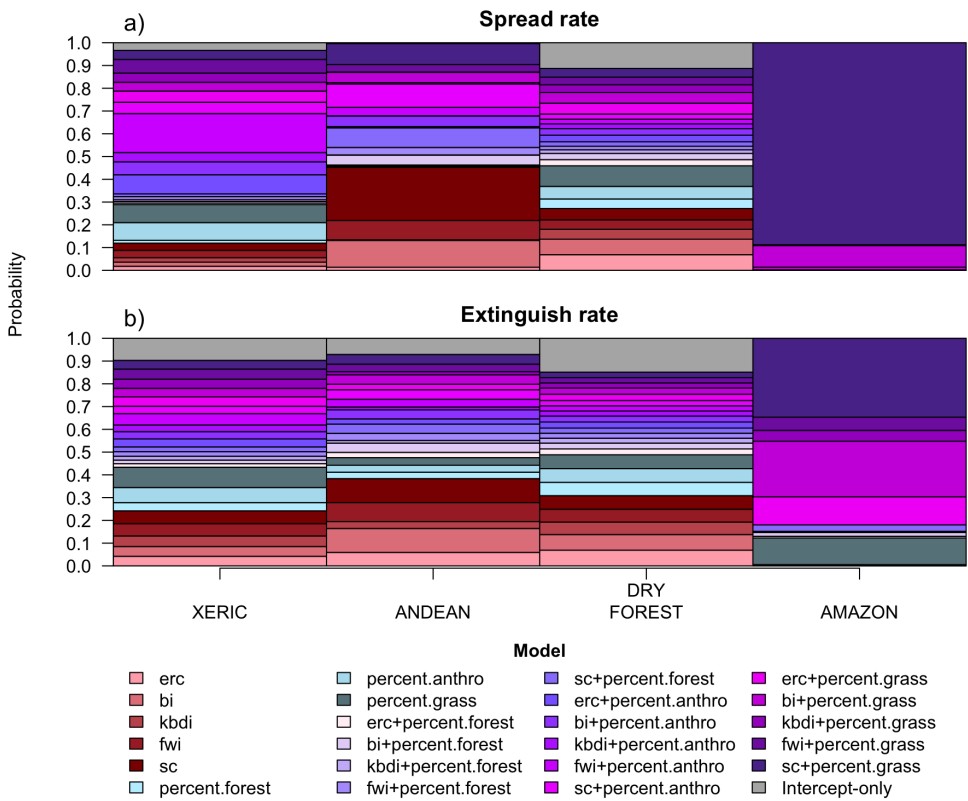

**Figure 6.** Barplots of Akaike weights - the probability that each model candidate is truly the best-approximating model - for each ecoregion and parameter. Models that contain both a FDI and land cover covariate are colored purple, those that contain only a FDI are colored red, those that only contain a land cover variable are colored blue, and the Intercept-only model are colored grey.

in RMSE estimates was not high. Similar results were observed in terms of relative performance. The spread rate estimates produced from the best-approximating generalized linear model were expected to get within 40 to 60 percent of the true spread rates, and when considering the sMAPE, the best model performance was again observed in the Amazon ecoregion, and the worst model performance was observed in the dry forest ecoregion (Table 5)

Model performance was slightly less in the extinguish rate models than in the spread rate models. The best-approximating generalized linear models were expected to get within about 1.3 to 2.6 of the best fitting spread rate estimates. The best model performance was observed in the Amazon ecoregion, and the worst model performance was observed in the dry forest ecoregion. Similar patterns were observed when considering the sMAPE. The best-approximating generalized linear models were expected to get within 70 to 100 percent of the true spread rates, with the best performance being observed in the Amaazon

ecoregion and the worst performance being observed in the dry forest ecoregion (Table 6).

    The effects of fire weather and land cover varied dramatically across ecoregions. In the xeric ecoregion, large FWI values were characterized with noticeably larger, faster growing, and longer duration fires compared to small FWI values. High

**Table 5.** Estimated performance of spread rate models as calculated from 3-fold cross validation. Each row reports the test years defining each fold.

| | XERIC | | | ANDEAN | | | DRY FOREST | | | AMAZON | | |
|---|---|---|---|---|---|---|---|---|---|---|---|---|
| | RMSE | sMAPE | n | RMSE | sMAPE | n | RMSE | sMAPE | n | RMSE | sMAPE | n |
| 2000-2006 | 0.23 | 52.70 | 5 | 0.30 | 54.33 | 54 | 0.32 | 46.69 | 8 | 0.21 | 36.45 | 170 |
| 2007-2013 | 0.22 | 31.94 | 7 | 0.32 | 43.81 | 73 | 0.42 | 78.88 | 8 | 0.22 | 44.70 | 225 |
| 2014-2020 | 0.48 | 50.61 | 26 | 0.39 | 44.62 | 126 | 0.45 | 47.93 | 34 | 0.38 | 46.53 | 268 |
| Average | 0.31 | 45.08 | | 0.34 | 47.59 | | 0.40 | 57.83 | | 0.27 | 42.56 | |

**Table 6.** Estimated performance of extinguish rate models as calculated from 3-fold cross validation. Each row reports the test years defining each fold.

| | XERIC | | | ANDEAN | | | DRY FOREST | | | AMAZON | | |
|---|---|---|---|---|---|---|---|---|---|---|---|---|
| | RMSE | sMAPE | n | RMSE | sMAPE | n | RMSE | sMAPE | n | RMSE | sMAPE | n |
| 2000-2006 | 1.19 | 88.99 | 5 | 1.58 | 100.36 | 54 | 2.26 | 80.25 | 8 | 1.00 | 59.66 | 170 |
| 2007-2013 | 1.39 | 56.32 | 7 | 1.82 | 77.12 | 73 | 2.87 | 131.29 | 8 | 1.12 | 80.44 | 225 |
| 2014-2020 | 2.71 | 78.65 | 26 | 2.31 | 79.72 | 126 | 2.56 | 92.56 | 34 | 1.66 | 75.28 | 268 |
| Average | 1.77 | 74.65 | | 1.90 | 85.73 | | 2.56 | 101.37 | | 1.26 | 71.79 | |

anthropogenic cover was associated with much smaller, slower spreading, and shorter duration fires compared to areas with low anthropogenic cover. Models that included fire weather and land cover had no detectable advantage over models that used no covariates at all in the dry forest ecoregions. In the Andean ecoregion, increases in the SC tended to result in slightly larger and faster growing fires, but the predicted fire duration was nominally smaller in the highest fire danger scenario considered. In the Amazon ecoregion, high SC values were characterized by increases in expected fire size and spread rate, and slight decreases in fire duration. Increases in grassland cover were predicted to increase fire size and spread rates, and decrease fire duration (Figure 7).

## 4.3 Sensitivity analysis

Repeating the analysis without filtering could change the model structure and uncertainty for both the spread and extinguish models. When the size-based filtering was omitted from the model fitting procedure in the xeric ecoregion, the best-approximating spread model changed from one that used FWI and anthropogenic landcover as covariates to one that used the SC and grassland cover as covariates. In the Andean ecoregion, the same changes in methodology changed the best-approximating spread model from one the used the SC only to one that used the BI and grassland cover as covariates. In the dry forest, these changes in methodology resulted in a best approximating spread model that used the burning index when previously no covariates were used in the best-approximating spread model. The most modest changes were observed in the

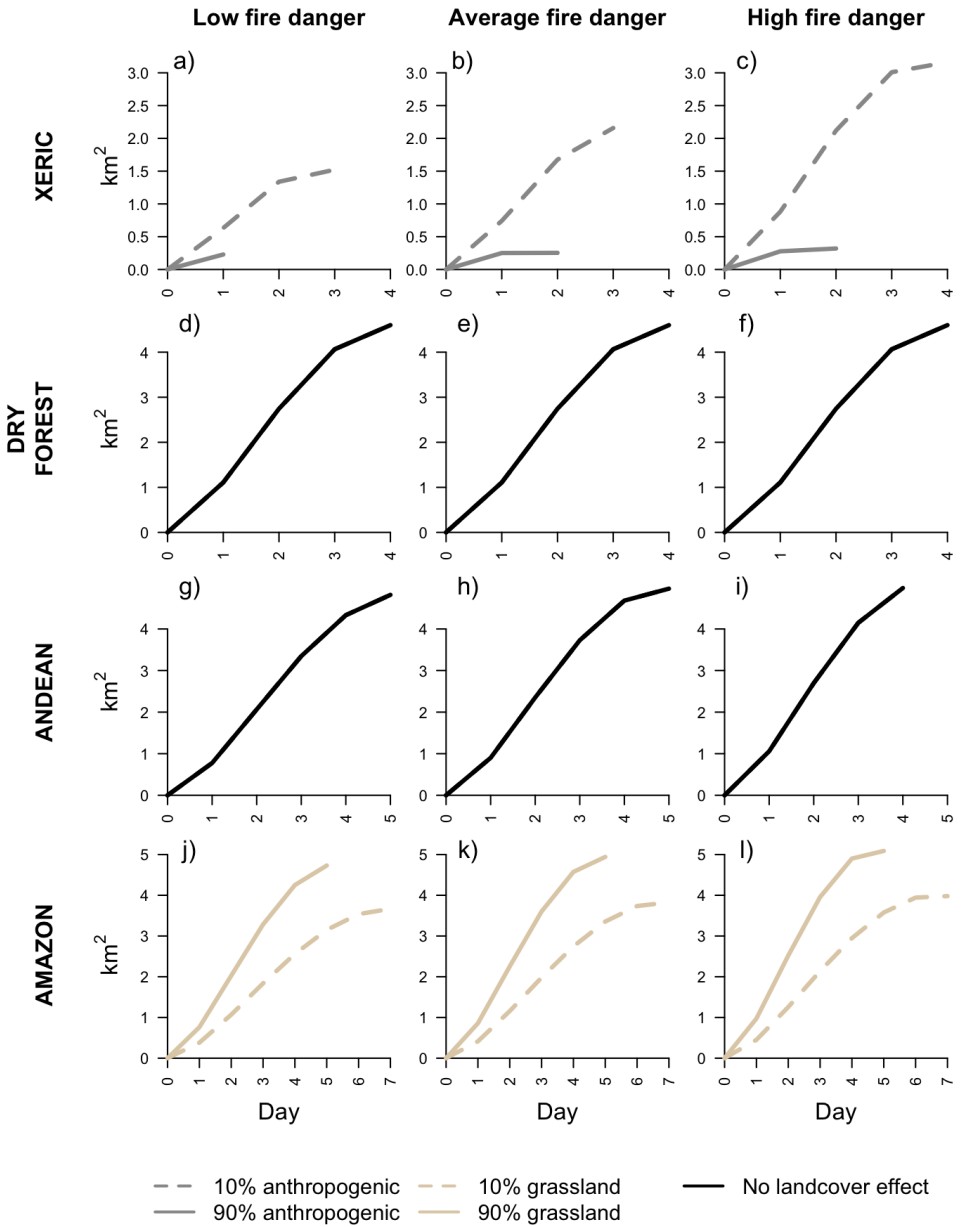

**Figure 7.** Expected daily burned area time series for each ecoregion under low fire danger (20th percentile of relevant FDI), normal fire danger (median value of the relevant FDI), and high fire danger (80th percentile of relevant FDI) conditions, and at two extremes of the relevant land cover covariate.

Amazon ecoregion, where the same set of covariates were used regardless if filtering was applied or not, although the effect that these covariates had on spread rates were sensitive to these changes in methodology (Table 7).

Similarly, the structure of the extinguish rate models were also sensitive to the choice to apply size-based filtering. When the size-based filtering was applied, the best-approximating extinguish rate model in xeric and dry forest ecoregions used no covariates, but when the size-based filtering was not applied, the best-approximating models used the SC and BI, respectively. As with the spread models, in the Andean region the best-approximating model used the spread component when size-based filtering was used, and used the BI and grassland cover when size-based filtering was not used. Lastly, the robustness to
choices in filtering methodology that was observed in the spread models for the Amazon ecoregion were likewise observed when considering the extinguish rate models (Table 8).

   Not only could the model structure change based on whether or not size-based filtering was applied, but the level of model uncertainty could also be sensitive to this decision. Specifically, AIC weights of the best-approximating models tended to be much higher when the size-based filtering was not applied compared to when it was. In other words, the probability that the
270 model that was identified as best, truly was the best-approximating model, was higher when size-based filtering was not used. The only exception to this trend was observed in the extinguish rate models for the dry forest ecoregion, where the AIC weight was 0.15 when size-based filtering was applied versus 0.11 when size-based filtering was not applied (Table 7-Table 8).

**Table 7.** Summary of spread parameter generalized linear models for each ecoregion as fit without using size-based filtering. Summary including formula, Akaike weight, and significance level. P-values greater than 0.10 are interpreted as not significant (ns), P-values less than 0.1 are interpreted as weakly significant (.), P-values less than 0.05 are interpreted as significant (\*), P-values less than 0.01 are interpreted as strongly significant (\*\*), and P-values less than 0.001 are interpreted as very-strongly significant (\*\*\*).

| Ecoregion | $-\frac{1}{\mu}$ | Dispersion | AIC weight | P-value | |
|---|---|---|---|---|---|
| | | | | FDI | Land cover |
| XERIC | $-3.76 + 0.36 \times SC + 0.47 \times percent.grass$ | 0.36 | 0.42 | \*\*\* | ns |
| ANDEAN | $-3.49 + 0.26 \times BI + 0.67 \times percent.grass$ | 0.37 | 0.97 | \*\*\* | \*\*\* |
| DRY FOREST | $-3.17 + 0.28 \times BI$ | 0.54 | 0.26 | \*\* | |
| AMAZON | $-3.69 + 0.12 \times SC + 0.86 \times percent.grass$ | 0.29 | 0.98 | \*\*\* | \*\*\* |

**Table 8.** Summary of extinguish parameter generalized linear models for each ecoregion as fit without using size-based filtering. Summary including formula, Akaike weight, and significance level. P-values greater than 0.10 are interpreted as not significant (ns), P-values less than 0.1 are interpreted as weakly significant (.), P-values less than 0.05 are interpreted as significant (\*), P-values less than 0.01 are interpreted as strongly significant (\*\*), and P-values less than 0.001 are interpreted as very-strongly significant (\*\*\*).

| Ecoregion | $-\frac{1}{\mu}$ | Dispersion | AIC weight | P-value | |
|---|---|---|---|---|---|
| | | | | FDI | Land cover |
| XERIC | $-1.01 + 0.13 \times SC$ | 1.10 | 0.15 | \*\* | |
| ANDEAN | $-1.01 + 0.10 \times BI + 0.20 \times percent.grass$ | 0.97 | 0.30 | \*\*\* | \*\* |
| DRY FOREST | $-0.96 + 0.12 \times BI$ | 1.59 | 0.11 | \* | |
| AMAZON | $-1.04 + 0.06 \times SC + 0.19 \times percent.grass$ | 0.91 | 0.57 | \*\*\* | \*\*\* |

## 5 Discussion

### 5.1 Difference equation validity

All models are approximations of the real world, and the relevant question to ask is whether the models can provide useful information (Anderson and Burnham, 2004). With this in mind, we argue that the difference equation model presented in this paper has a number of advantages over other approaches used to model fire growth. Firstly, this model is simple, requiring estimation of only a few parameters. This model is therefore vastly easier to implement than other commonly used spread models that require complex data inputs and subjective decisions regarding initialization parameters (Sullivan, 2009b, a). The

model parameters are also easily interpreted and are directly related to the underlying fire processes. Although alternative difference equation models exist that could adequately represent observed fire growth (Podschwit et al., 2018), the interpretation of the model parameters in these alternative models is cumbersome because they were not originally developed to represent fire growth.

Most importantly, these difference equations well-approximated observed fire growth data (Figure 5). Although the RMSE

deteriorated in larger, longer-duration fires, the relative performance was quite good (Figure 5) and is a tendency that is not unique to this model in any event (Podschwit et al., 2018). The fact that the spatial distribution of predicted spread rates conformed to previously known patterns of fire spread adds to the credibility of these models. Specifically, that the lowest spread parameter rates were observed in the Amazon ecoregion is largely consistent with existing research (Ray et al., 2005; Cochrane, 2003; Andela et al., 2019), as is the fact that faster spread rates were observed in ecoregions that are largely non-forested

(Massman et al., 2017; Andela et al., 2019). Because burned area estimates were derived from satellite instrumentation, some unburned islands were inevitably treated as burned (Kolden and Weisberg, 2007), and as is commonly reported in other data (Short, 2014), GlobFire data were observed to sometimes classify spatially distinct fire events as a single fire. In aggregate, these biases should result in models that overestimate individual fire spread rates, however, the mean spread estimates from the Amazon ecoregion at least (0.44 km/day) did not dramatically differ from published ground-based estimates (Ray et al., 2005).

It should be noted that the model parameters for each fire were estimated as to minimize the RMSE and did not account for other measures of goodness-of-fit. Consequently, the modeled growth curve sometimes extended beyond the observed duration of fire, so that although the growth curve matched the observations well, the estimates of eventual burned area and duration from the model were too large. This fact is important to keep in mind when interpreting the slight decrease in duration associated with high FDI values in the Amazon and Andean ecoregions (Figure 7). Because of these occasional counterintuitive predictions,

we recommend that these models primarily be used to estimate spread rates, and that predictions of duration and eventual burned area be used with caution. The estimates of spread rates could be useful proxies for burnover risk and suppression effort for most ecoregions, but the Amazon may be somewhat of an exception. In the Amazon, the slow fire spread rates, low flame heights (Cochrane, 2003; Ray et al., 2005; Bufacchi et al., 2017), and sparse settlement suggest that explosive fire growth poses a practically non-existent threat to humans. Still, the Amazon should only very rarely have naturally occurring fire (Lima et al.,

2012; Cochrane, 2003), and the spread model predictions might find use as a tool to identify when and where fire occurrence is at all possible.

## 5.2 Effects of meteorology and land cover on fire development

That fire spread was generally predicted to be higher during meteorological conditions considered conducive to fire spread is expected. And although the identification of the SC - which is interpreted as an estimate of idealized fire spread (Bradshaw et al., 1984) - as the best FDI in the Andean ecoregion is not surprising given that the ecoregion is generally non-forested, the fact that this same relationship was observed in the Amazon ecoregion was noteworthy. One might not predict that a FDI that was largely related to wind would be a useful proxy for fire spread in heavily forested and tropical ecosystems, where a FDI that instead measured sustained drought might be expected to perform better (Cochrane, 2003). However, the SC is not only computed from wind but also from the moisture content of the smaller fuel size classes (Bradshaw et al., 1984) and it is therefore plausible that the SC was selected because, unlike the BI or ERC, it was a proxy of fuel moisture of lighter fuels that are characteristic of the Amazon's understory litter layer (Cochrane, 2003). These results also suggest that decision makers should reconsider their preferential use of other FDIs over the SC (Jolly et al., 2019), as the latter was the apparent best predictor of fire spread in multiple ecoregions (Table 3-Table 4). The lack of a strong relationship between FDIs and fire spread in the dry forest and the weak relationship in the xeric ecoregion may also seem surprising. One explanation for this apparent lack of correlation in these two ecoregions is that the elevated human habitation in these locations relative to other portions of Peru reduces the effects of meteorology on fire spread, which is a phenomenon that has been observed in the United States (Syphard et al., 2017). Indeed, in the xeric ecoregion, anthropogenic cover had a weakly significant negative effect on fire spread (Table 3), suggesting that nearby human presence may inhibit wildfire activity. Anthropogenic effects were less plausible explanatory variables in the Andean and Amazon ecoregions, which have relatively sparse human habitation[5], and the covariates used in the best-approximating models of these ecoregions reflect this fact (Table 3, Table 4). Another explanation might simply be that the small sample size did not permit the selection of complicated models, and in fact when size-based filtering was not performed, the BI was observed to have a significant and positive effect on fire spread in the dry forests. Grassland cover was a positive and very-strongly significant predictor of fire spread in the Amazon, which, as mentioned in the previous subsection, conforms with existing research regarding the global variability in fire spread rates (Andela et al., 2019), as well as research regarding the effects of vegetation of fire spread (Massman et al., 2017). Overall the results from the original model fitting and sensitivity analysis suggest that, in general, fire spread is enhanced during dry windy conditions and can sometimes be exaggerated in grassland environments ((Table 3, Table 7)).

In ecoregions where covariates were relevant, the predicted relationships between the extinguish rates and the environment were sometimes counterintuitive. For instance, in the Andean and Amazon ecoregions, extinguish rates were positively related to FDIs, suggesting that drier conditions were conducive to fire extinguishment. The relationships between fire danger and extinguish rates are not easily dismissed as they are strongly significant (Table 4), and in the Amazon at least, we have fairly high confidence that the model is the best out of the 24 candidates (Figure 6). However, a satisfactory explanation for this counterintuitive result can be arrived at by (1) revisiting the original mathematical model, (2) observing the correlations between the two parameters, and (3) understanding that the counter-intuitive results may arise from constraints imposed by the model

---

[5]https://sedac.ciesin.columbia.edu/data/set/grump-v1-population-density/maps?facets=region:south%20america

structure rather than real-world relationships. Consider that, as described at the beginning of subsection 2.3, the duration of the fire is controlled by the relative decay rate, which is the ratio of the spread rate and extinguish rate. This implies that the spread rate is approximately proportional to the extinguish rate, with a harmonic number as the scalar. We further note here that the harmonic number scalar is a function of the fire's duration. We can then envision two ways that high covariance between the spread rates and extinguish rates might arise. First, if the variability in the duration is not too large, then the harmonic number scalars will not strongly vary, and spread rates will be approximately proportional to extinguish rates. Second, if the average fire duration is large enough, then the harmonic number scalars will also not strongly vary. In any event, the mean and variance of the fire duration distributions resulted in spread rates and extinguish rates with high covariance, as is evidenced from the high spearman correlation coefficients and the strong fit of the quadratic models (Figure 4). It is then not surprising that similar models would independently arise when estimating model fit from two parameters that are so closely correlated. Indeed, in the Amazon ecoregion, we can see that the exact same covariates are used in the extinguish rate model and the spread rate model (Table 3, Table 4) and that the direction of these effects are the same. Because the same covariates are unlikely to be sufficient explanations for the two contradictory processes of fire spread and fire extinguishment, we will note here that we believe that although some of the relationships observed here reflect real-world, others are likely the result of constraints created from the difference equation model. Specifically, the fact that spread rates are positively correlated with high fire danger and grassland cover is both consistent with (1) ones intuition about how the environment should interact with fire and (2) existing research in other locations. On the other hand, it does not seem likely that conditions that enhance fire spread also hasten fire extinguishment, and we think that the relationships between extinguish rates and the environment are better explained as model artifacts.

In aggregate, the spread rate and extinguish rate models produce fire progressions that are plausible. In all but the dry forest region, increases in FDIs produce larger and faster growing fires. Moreover, the effects of landcover are likewise believable. In the xeric ecoregion it would make sense that - given the destructive effects of fire to human health and economic activity (Stephens et al., 2014) - human presence would have an inhibitory effect on fire. Similarly, given that forest vegetation can inhibit fire growth (Massman et al., 2017) it is reasonable to predict larger and faster growing fires in grassland cover. The negative relationship between FDIs and duration should not be accepted uncritically, although potential explanations could exist. In fragmented landscapes, it is conceivable that fire size is constrained by available fuels and that increased fire danger merely hastens the consumption of these fuels. That is, fire size may be fixed due to fuel discontinuity, and increased fire danger just hastens the inevitable in these circumstances. However, fragmentation can be an unsatisfactory explanation for much of Peru, and like the relationship between extinguish rates and FDIs, we should also not discount the possibility that the predicted relationship between fire duration and FDIs might be better explained as a model artifact rather than a real-world causal relationship.

## 5.3 Future work

The results and methods in this paper provide a template for forecasting fire growth from environmental covariates. When a fire ignition is reported, estimates of the spread rate and extinguish rate can be calculated from environmental data, which would

permit estimates of risk. Because the spatial domain of the GlobFire data covers the entire world, the methods described here can be readily applied to other ecoregions, which would be particularly useful to locations without access to sophisticated fire spread modeling capabilities. Although predictions from these models describe a time series of expected fire growth conditional on meteorological and land cover conditions, it is often the statistical extremes that are the most destructive and dangerous (Stephens et al., 2014; Viegas and Simeoni, 2011). These extreme events often occur in unique conditions (Slocum et al., 2010; Barbero et al., 2015a) and there relative position in the statistical distribution of fire sizes means that these events are some of the hardest to predict (Figure 5). For this reason, future work should further validate the predictive performance of these models to determine what additional information and methods are required to best predict the times when these extreme fires occur.

Future work might include additional covariates into these models to further improve predictive performance and by extension improve fire-related decisions. In addition to those explored in this paper, there are numerous other FDIs that could be included that might better model fire spread (Littell et al., 2016; Meyn et al., 2007; Barbero et al., 2015b). Firefighting-related covariates could be used to forecast the effects various suppression strategies would have on fire behavior, and various ocean temperature indexes could be included to account for ENSO-related effects on weather that modulate fire spread (Chen et al., 2011; Block et al., 2000). Forest structure and composition is a highly complex and important variable to consider when estimating fire growth, but was not thoroughly investigated in this analysis for practical purposes. A South American forest containing native species versus one that is dominated by eucalyptus (Galizia et al., 2021) might burn much differently even though both could be characterized as forested (Cruz et al., 2021). Projected climatic and land use changes could be used to provide longer-term forecasts of fire risk and inform decisions such as firefighting staffing and training programs, as well as inform land management decisions. The spread rates and extinguish rates would clearly differ between planned and unplanned fires, but differentiating between these two types of fires was not attempted in this study due to the difficulty in making this categorization from satellite data.

In addition to exploring additional covariates, the difference equation could be modified to better describe the complexity of fire progression. For instance, the assumption of uniform growth in all directions is an oversimplification, and it is worth exploring what performance improvements might arise from permitting different spread rates at the head and flank of the fire. We will note here that such an approach would immediately create at least two issues. Firstly, such a model would require the estimation of the partial perimeter of an ellipse, which does not have a convenient closed-form solution like is seen when circular fire growth is assumed. This could be solved numerically, but a second issue arises from uncertainty about which segment of the ellipses burning perimeter is extinguished at each time step. That is, the progression of a fire in which the head fire is extinguished first may be much different from one in which the flanks are first extinguished. One potentially acceptable solution to this problem might be considering a best and worst case progression scenario. The majority of fire spread typically occurs over only a few days of a fire (Wang et al., 2017), and therefore the day-to-day variation in fire spread is an important variable for assessing burnover risk and, by extension, firefighting decision-making. Other parameterizations might then add greater realism by allowing spread rates or extinguish rates to change over a fire's lifetime, or by using daily variation in certain covariates to explain day-to-day changes in spread rates or extinguish rates. Although future investigations might successfully

overcome these challenges, we quickly found that this task presented serious computational difficulties that were avoided with the simpler methods described in this paper.

Given the high covariance of the two difference equation parameters and sometimes counterintuitive results, future work should also explore the necessity of forecasting extinguish rates from environmental data or if the extinguish rates may simply be inferred from predicted spread rates instead (Figure 4). The model inputs are subject to uncertainty and future work might investigate what the consequences of this uncertainty might be. For example, land cover estimates change over time, and areas that were once forested may become unforested, and vice-versa. Hence, it is possible that some of the land cover data

are corrupted and crosschecking the results using other datasets can ensure that the results are robust. Similarly, although using meteorological information at the ignition date was a methodological convenience, it may not be representative of the conditions that occurred during the largest growth days. Although we determined that the level of temporal autocorrelation in the FDIs was reasonably high enough to justify using data at the ignition date, this uncertainty in the data inputs should be investigated further. We would like to note that performance was best in the the Amazon, where most of the data was available,

and performance was most variable in the ecoregions with the least data. We further note here that more complex models were permitted when size-based filtering was not performed. Both of these result suggest a need for continued data collection, so that sufficient amounts of data are available to produce reliable estimates of performance, and that important relationships between fire and the environment are able to be identified.

## 6   Conclusions

In this paper, we developed a simple and intuitive difference equation model of fire growth that can be estimated using globally available satellite data. The difference equation was based on two parameters: the spread rate and the extinguish rate. We described the statistical distribution of these two parameters for four ecoregions in Peru using Gamma distributions. We also built generalized linear models to predict these parameters using FDI and land cover as covariates. We found that FDIs - specifically SC, FWI, and BI - were useful predictors of fire spread and extinguishment in Peru. In general, when FDIs were

used to predict these two parameters, we found that the SC was usually the best choice. However, model uncertainty was frequently high and the identification of which FDI was best was sensitive to methodological choices regarding size-based filtering of fires. Compared to other regions, model confidence was particularly high in the Amazon ecoregion and low in the dry forest ecoregion. In addition to FDIs, land cover was also a useful predictor in some contexts. Specifically, anthropogenic cover had a weakly significant and negative effect on spread rates in the xeric ecoregion, and grassland cover had a postive and

often very-strongly significant effect on spread and extinguish rates in several contexts. Counterintuitive relationships were observed between extinguish rates and FDIs, where increased fire danger increased extinguish rates. These relationships were explained by the fairly strong correlation of the two model parameters and were concluded to be model artifacts rather than representing real-world causal relationships. We argue that the methods presented here, although simple and can be improved upon, are a promising method of forecasting fire spread risk when sophisticated fire modeling capabilities are unavailable or

impractical.

*Author contributions.* conceptualization, H.R.P.; methodology, H.R.P. and W.M.J; software, H.R.P. and A.M.; validation, H.R.P. and W.M.J; formal analysis, H.R.P. and A.M.; investigation, H.R.P.; resources, E.A., and W.M.J; data curation, W.M.J and A.M.; writing–original draft preparation, H.R.P.; writing–review and editing, H.R.P., W.M.J, E.A., A.M., S.V., S.B.R., C.T.C., and B.P.V.; visualization, H.R.P. and S.B.R.; supervision, W.M.J and E.A.; project administration, H.R.P, B.P.V., W.M.J, and E.A.; funding acquisition, W.M.J and E.A.

*Competing interests.* The authors declare no competing interests.

*Acknowledgements.* This work was funded in part by the US Forest Service International Programs, the United States Agency for International Development, USFS Rocky Mountain Research Station, University of Washington, and the Fulbright Scholar Program. The authors would like to thank Isidoro Solis, Vannia Aliaga-Nestares, and Diego Rodriguez-Zimmermann.

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

**Appendix A: Q-Q plots**

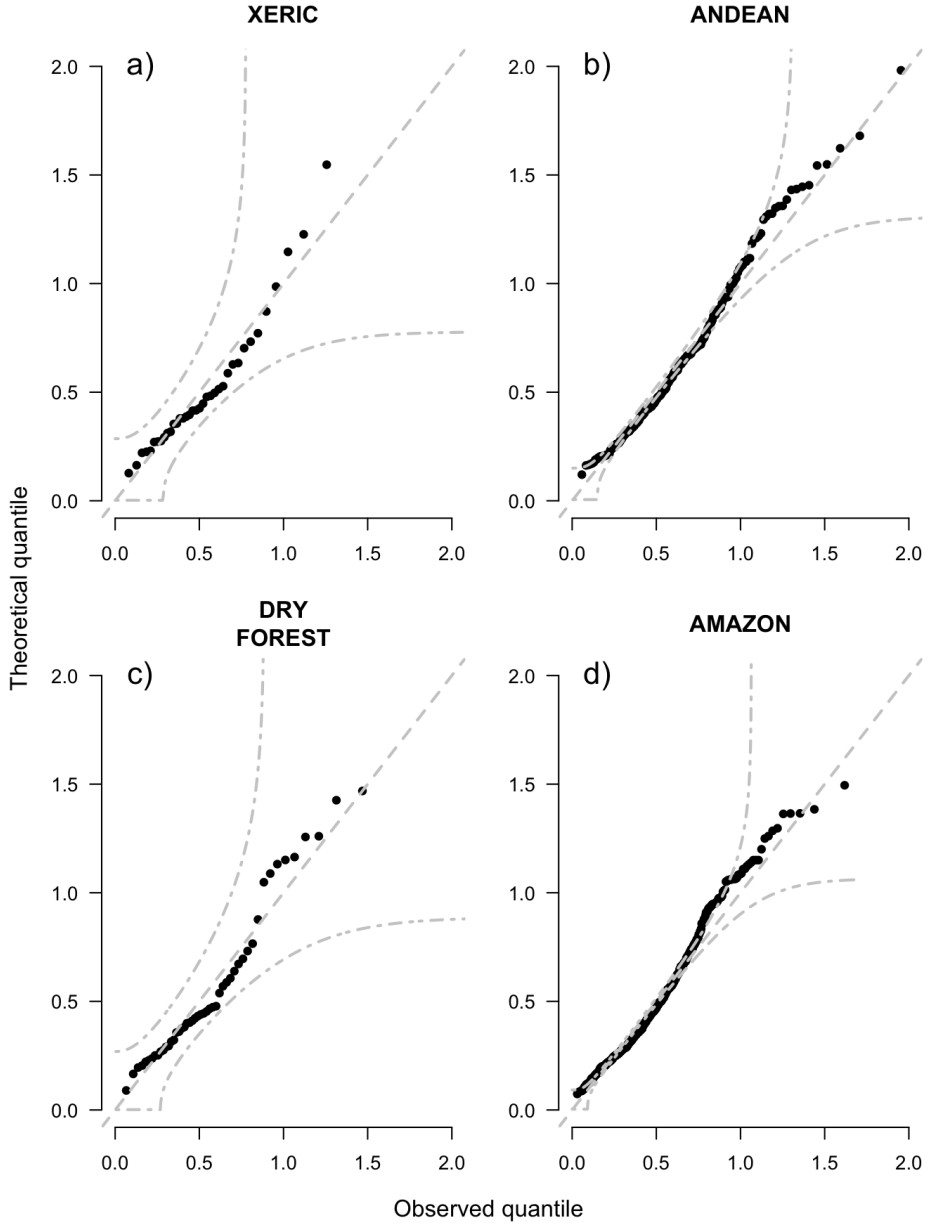

**Figure A1.** Q-Q plots of spread rate parameters disaggregated by ecoregion. 95% and median simultaneous confidence band shown in dashed lines.

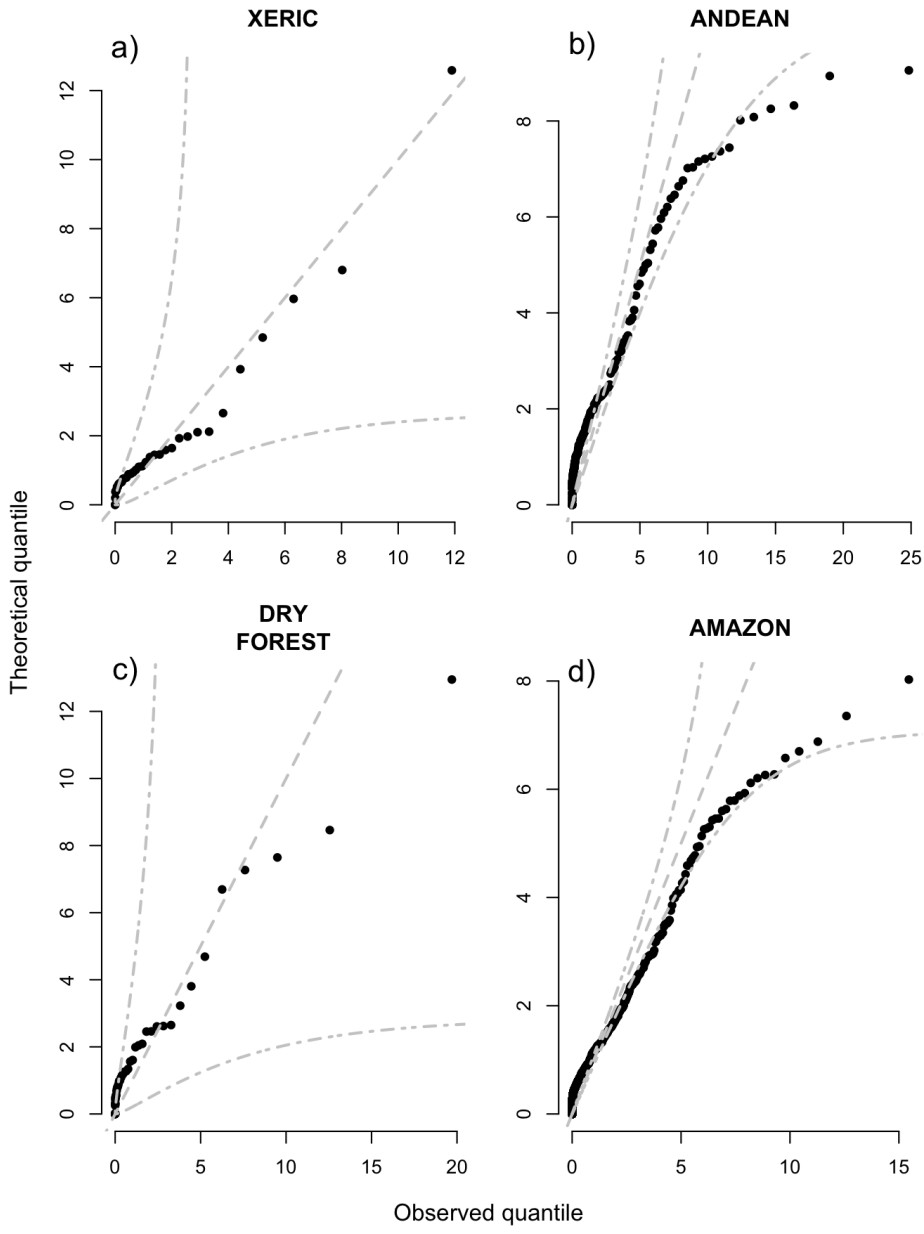

**Figure B1.** Q-Q plots of extinguish rate parameters disaggregated by ecoregion. 95% and median simultaneous confidence band shown in dashed lines.