# Peer review of "Estimating the effects of meteorology and land cover on fire growth in Peru using a novel difference equation model"

_EGUsphere, 2022_

## Author Response (AR1)

Reviewer 1

As mentioned in our previous response, we agreed with nearly all of reviewer 1's comments, found them helpful, and had already incorporated them into the manuscript during the previous response round.

General comments

This article by Podschwit et al. introduces a novel and simple method to model wildfire growth. Specifically, a difference equation model estimating a spread and an extinguish parameter was described and generalized linear models were fit for each parameter which use fire danger indexes and land cover predictors. The method was tested using fire perimeter data from recent wildfires in four ecoregions in Peru. The approach is certainly interesting and the methodology is mostly comprehensible, with a few key points needing to be further clarified. The overall presentation of the results is sound and the common thread can be followed throughout the paper while language and readability are almost flawless. Here and there, rework is needed to clarify certain points which currently might confuse readers. Therefore I propose that the article can be accepted for publication in this journal after addressing some minor revisions.

Specific comments

While I understand the intention and the setup of the study, I think the methodology currently lacks some clarity. Firstly, Figure 1 makes it seem like direction matters to the approach but if I understood correctly, the extinguish parameter merely decreases the perimeter as a whole and not on a specific side of the circle. Speaking of the sector arc makes this further confusing but I understand that it needs to be calculated for solving the difference equation. I think the best way to avoid this confusion is to adjust Figure 1 and have the "fire" spread in all directions instead of just one. Overall, I think it would still be good if there is some information about the direction of spread (e.g. that you are not modeling it here and why).

Figure 1 has been corrected and language has been included in the discussion that clarify reviewer 1's concerns regarding confusion about direction of spread.

At the end of the model description, it would be nice to also show the final difference equation (after Line 116). This might of course be trivial to some but I think others would appreciate to see the final equation directly and it would serve as a good end point of section 2.

This has been included.

The data selection and description is okay. Maybe it is a good idea to state which year the Nature Conservancy data for the ecoregion definition is from. Besides that, I wonder if it is possible to consider land cover changes throughout the study period. Linking a land cover map from 2009 to a fire event from 2019 could be problematic. I understand that the latest GlobCover dataset is only available for 2009 but other global datasets with higher spatial and temporal resolution might increase the validity of the analysis. I was also asking myself why the land cover data was only reclassified into two categorical values. I understand that it is a simple approach but this again raises the question why GlobCover was chosen specifically. At the very least, some more information about the land cover data should be provided (around Line 130).

We included a description of the GlobCover data, mentioned the year in which it was from, and also briefly discussed the implications of uncertainty in the model inputs in the 'Future Work' section.

One more parameter which can potentially also influence fire spread is forest structure (or forest composition). This seems to not be considered here. Is there a specific reason for that?

The topic of forest composition and structure is now explicitly mentioned in the Discussion section when the topic of additional covariates that were not considered in the original analysis is being discussed.

I would also suggest giving some more information as to how the elements of the GLM were chosen. It seems like an inverse-link and a Gamma density function work here and make sense but when reading the text, this assumption comes out of nowhere and it would increase the understanding of the modeling approach if this part was a bit more elaborated on.

Language was included to justify this decision and a footnote was included to let readers know that QQ-plots of the alternative models would be provided upon request.

To give the article a better structure, I suggest moving the results section into a separate chapter (e.g. a new chapter 4). Currently, the results are announced in chapter 3.4 but then the actual results follow in chapters 3.5-3.7. It would make much more sense to put them into a new chapter to separate them from the data and modeling section.

On a similar note, I suggest getting rid of chapter 3.7 and work the contents into chapter 3.6 where the same relationships are already discussed. Figure 7 should be kept and the growth curves should be discussed more thoroughly.

Done.

I think the discussion part is okay and covers the interpretation of results, potential flaws of the approach and possible future work. However, what I'm most concerned with is the explanation for the counterintuitive results of the relationships between the extinguish rates and the environmental variables. Your explanation sounds like this comes solely from the correlation with the spread rates. This sounds correct but it would mean that the extinguish parameter is not independently modeled and is therefore flawed. It would be great to get some more insights as to why the analysis was still carried through with this approach.

This was also a bug bear of one of the other reviewers, in addition to the explanation that was provided in the previous response, the language regarding the counter-intuitive results was clarified. Specifically, it is mentioned that the extinguish rate results are likely a model artifact rather than a real-world, causal relationship and there is little intuition or academic research justifying believing otherwise.

Finally, I'd like to make a general remark about the title and the contents of the article. The title made me expect an application of known methods to a specific study region. However, while reading it felt more like a methodological article which describes and evaluates a novel and simple approach for modeling fire growth. Maybe the title can be adjusted to accurately state what the key aim of this study was.

The title has been changed.

Reviewer 2

The manuscript provides a simple wildfire growth model for four ecoregions in Peru. The model relates fire growth to fire danger indices and land cover through difference equation models that are parameterized to best fit 1003 large multi-day fires in terms of radial spread speed and perimeter length extinguish rates. The differences found for Andean (grass), Xeric, Dry Forest and Amazon Forest are calculated and discussed. Potential applications are alluded to.

General comments:

1. The manuscript is well written and clearly discusses the materials.
2. The method and calculations are well explained.
3. As constructed, the validity of the approach is uncertain since no validation was attempted.

A 3-fold cross-validation was included in the updated manuscript.

4. Be careful in discerning where your model is giving you insights into fire behavior as opposed to where results are artifacts of the model constraints or assumptions.

This was explicitly addressed in the revised manuscript. Specifically, the strange results regarding the extinguish rates and the FDIs was concluded to be a model artifact, as there is very little intuition or academic research to justify a belief that FDIs make fires extinguish faster.

Specific comments:

1. The model makes two simplifying assumptions, namely that fire spreads at a constant rate from an ignition point, and that a constant length of the fire's perimeter is extinguished after the first time step. This is an understandable expediency but how and when these assumptions might lead to erroneous results should be discussed. Basically, what sort of fire behavior or conditions would tend to 'break' this model?

In the future work section, we discuss how the model is an oversimplification and how future researchers can include additional complexity to make more realistic predictions.

2. Provide more detail on the burned area product being used. The GlobFire data is derived from a 500m MODIS burned area product.

This has been elaborated upon in the revised manuscript when the data are introduced.

3. Many fires over many years were used to derive the specific ecoregion models, which is good, but effectively what is developed are average values for each fire. Since daily spread is not examined, it would seem that basically, the final Area (km2) and Duration (days) are used to solve for an average spread rate (r) and fixed amount of perimeter being extinguished. The lack of daily progressions is clearly discussed in the Discussion section but more could be done to 1) support the approach, and 2) explain why it may not work so well for some ecoregions.
    1. Why were all the fires used in the parameterization? Generally it would be expected to break into training and validation datasets. This division could be done and analyzed in a number of ways to better understand how well the model(s) are performing. Equally well across fire danger levels? Equally well for each year?
    2. How was the fire danger that was used calculated? From the text, it states that the value on the date and location of the reported ignition was used? Did you look at how much daily variability there was over at least a selection of the fires in each ecoregion? Given the approach, the median or average value for the duration would seem more likely to prove significant. This may be part of the reason why the model fit drops as the duration (and hence area) grows.

3. It should be noted that the model did fabulously in the Amazon (n = 663), reasonably in the Andean (n=252) and marginally/poorly in the Xeric and Dry Forest (n = 38 and 50). Could performance be sample size driven? That may not be the driving issue but it could be tested to bound the matter.

A 3-fold validation was included in the revised manuscript and a better explanation of the fire danger variables is now provided in the introduction. To the second point, we elaborated upon how covariates could be used to predict daily variability in fire spread (see the Future work section in the paragraph detailed how the difference equation parameterization could be modified). The closing sentence of the Future work section also explores the importance of data availability for reliable estimation of model performance and avoidance of underfitting.

4. Given the methods, I understand why single day fires were excluded but where does the 405 hectare limit come from? Is it reasonable to have it be the same in all ecoregions?

A sensitivity analysis is now included that explicitly repeats the analysis without the size-based filtering.

5. Line 162 - "The spread rate and extinguish rate were highly correlated with one another": Given that they were both calculated from the total area burned and duration of burning, is this a finding or a necessity from the calculations?

The correlation of extinguish rates and spread rates, and its relationship to the difference equation are thoroughly discussed in the "Effects of meteorology and land cover on fire development" section.

6. Do the median RMSE values for the ecoregions have any independent comparative value? Shouldn't they be normalized by the median spread rate or fire size for each ecoregion? It is unclear if 0.5 km2 is a large or small error, for example.

The (S)MAPE is included as a performance measure in the revised manuscript, which is a relative measure of performance. The higher model performance in smaller, shorter duration fires erases when the relative measure is used instead.

7. Figure 5 (Time – Days) – this looks like the error term is an exponential term. The long term rise in errors would drive the larger error for large fires and ecoregions typified by them. Then again it may be a sample size issue….

This is addressed by (1) the use of relative performance measures and (2) our discussion of the importance of continued data collection at the end of the "Future work" section

8. Line 191 – "Changes in fire weather and land cover were predicted to have no effects on fire growth in the dry forest regions."- Really? I think this is misstated or fundamentally miscomprehended. What was shown here is that neither fire weather nor land cover had statistically significant predictive value for fire growth in these forests. Why is the question? Is it because there were no other land cover types

encountered? Was it because fire growth was more strongly determined by a variable that wasn't included (e.g. topography)? Was it because of the relatively small sample size, especially since the data were skewed by at least one very large fire?

This sentence was replaced with a more precise wording, and the fact that this result arose are detailed in the closing sentences of the "Future work" section that outline the importance of continued data collection. We saw that these covariates WERE important when the size-based filtering was omitted.

9. Line 209 – additional parameters or covariates could be included" – such as what? How would they help?

The future work section has two paragraphs elaborating each of these aspects ( 1. additional covariates and 2. modifications to the difference equation).

10. Figure 7 – Wettest 2% of day? If that were true no fire spread would be expected. I suspect this is the wettest 2% of days when fire spread was observed.

The 20th and 80th percentile were used in the revised manuscript, which as mentioned in the previous response, represent the minimum percentile at which fire activity was observed (in all ecoregions).

11. Line 228 – How would the spread tool identify when/where fire occurrence is possible since it is not spatial? It might help identify conditions when fire spread is possible that could perhaps be used in this way.

12. Lines 244-245 – "relatively sparse human habitation" – perhaps but in the case of the Amazon forests most if not all of those ignitions are tied to those areas of sparse human habitation. The fires rarely if ever happen in remote forests.

We tried to use more precise language in these two passages.

13. Lines 256-257 "This implies that the spread rate is proportional to the extinguish rate, with a harmonic number as the scaler" – see comment 5 above.
14. Line 269 – "additional covariates" – such as?

Detailed in the relevant "Future work" section paragraph.

15. Line 274 – "ENSO-related effects on fire spread" – perhaps better stated as – "ENSO-related effects on weather conditions that affect fire spread".

This was corrected. Yes, ENSO changes the weather, which modulates fire activity.

16. Line 305 – "increased fire danger increased extinguish rates" seems nonsensical physically. It could arise from the model structure for fires that either were extinguished because they spread more rapidly to where landcover or features prevented further fire spread or when sudden events (e.g. rain) extinguished a fire that started during high fire danger conditions.

Reviewer 2 will find that this has been thoroughly addressed. We conclude that the weird relationship between extinguish rates and FDIs is more likely a model artifact rather than a real-world causal relationship.

Reviewer 3
This paper presents a proposal for a simple difference equation of fire growth based on the spread rate and extinguish rate, obtained from fire perimeters in Peru based on the GlobFire dataset. The authors present generalized linear models for four biomes in Peru using fire danger indices and land cover as covariates. The method is interesting, but it includes many assumptions that are not easy to accept, and that limit the capacity to extrapolate the results to other areas.

General comments:

The methods in Section 3 should be improved and clarified. Any reader not expert on fire behaviour will probably not be able to understand what the different FDIs are, which is the difference between them, the variables used to calculate them, and why they were selected amongst the different NFDRS components. Although some of this information is commented in the discussion, that is not the section to explain how the FDIs are calculated.

The revised version includes a brief description of the FDIs in introductory materials.

The same applies to the methods applied to estimate the model parameters and perform the statistical modelling. Further explanation should be provided to help the interpretation of the results. For example, I do not see a clear relation between the parameters "relative decay rate" and "normalizing factor" in Section 2, and the methods explained in Section 3.

We included a brief passage to make it clear that the relative decay rate controls the fire duration. The normalizing factor is used to take the relative decay rate, which controls fire duration, and estimate a spread rate and extinguish rate the produces a burned area time series that reaches the desired final area. The critical piece that reduced the confusion was by reminding readers of Equation 11.

I have some concerns regarding the analysis performed:

There is a group of land covers not considered in the analysis, corresponding to the shrubland and grassland land covers. I believe that it is an important omission.

It would be expected that in the Andean region (which actual biome name is "Montane Grassland and Shrublands) fires would be influenced by these land covers, but that is not reflected in the paper.
It is difficult to accept that any good model trying to characterize fire behaviour would be independent from fuel characteristics and meteorological conditions.
It is a very risky assumption to use the FDIs calculated at one point in time (on fire start), and with a very coarse resolution, as characteristic of a fire event that had a size of several km2, and spread during days or weeks. Although the authors mention it in the discussion, I believe that as is, the model is very limited in this respect.

Grassland landcover was included as a covariate in this manuscript and it turned out to be more important than forest or anthropogenic cover.

We mentioned that we believed that the level of temporal autocorrelation was sufficiently high to make reasonable predictions, but that the uncertainty in the meteorological inputs should be examined by future researchers in the future work section.

Although the authors state that estimates of spread and extinguish rate could be calculated from environmental data, Tables 3-4 suggest the opposite for the biomes where FDIs are not included in the linear models.

The sensitivity analysis that is now included in the manuscript makes clear that FDIs and land cover are definitely plausible predictors of the difference equation parameters, even though they were not in the best-approximating models when size-based filtering was applied. Moreover, as mentioned in the future work section - specifically the additional covariates paragraph - other environmental variables (beyond those explored in this manuscript) *could* be important predictors of the difference equation parameters.

From the reading of the paper I understand that all the Peruvian fires that complied with the size and time length distribution were used to calibrate the models. I would have liked to see some events left out to be used as validation of the model, in order to evaluate its applicability to other fire events.

A 3-fold cross validation was included in the revised manuscript.

I agree with the authors in their statement that more focus should be given to extreme weather conditions, since they are the ones that probably caused the large fire events used in the paper.